# DIFFERENTIALLY PRIVATE SYNTHETIC DATA: APPLIED EVALUATIONS AND ENHANCEMENTS

## ABSTRACT

Machine learning practitioners frequently seek to leverage the most informative available data, without violating the data owner's privacy, when building predictive models. Differentially private data synthesis protects personal details from exposure, and allows for the training of differentially private machine learning models on privately generated datasets. But how can we effectively assess the efficacy of differentially private synthetic data? In this paper, we survey four differentially private generative adversarial networks for data synthesis. We evaluate each of them at scale on five standard tabular datasets, and in two applied industry scenarios. We benchmark with novel metrics from recent literature and other standard machine learning tools. Our results suggest some synthesizers are more applicable for different privacy budgets, and we further demonstrate complicating domain-based tradeoffs in selecting an approach. We offer experimental learning on applied machine learning scenarios with private internal data to researchers and practitioners alike. In addition, we propose QUAIL, a two model hybrid approach to generating synthetic data. We examine QUAIL's tradeoffs, and note circumstances in which it outperforms baseline differentially private supervised learning models under the same budget constraint.

## 1 INTRODUCTION

Maintaining an individual's privacy is a major concern when collecting sensitive information from groups or organizations. A formalization of privacy, known as differential privacy, has become the gold standard with which to protect information from malicious agents (Dwork, TAMC 2008). Differential privacy offers some of the most stringent known theoretical privacy guarantees (Dwork et al., 2014). Intuitively, for some query on some dataset, a differentially private algorithm produces an output, regulated by a privacy parameter $\epsilon$, that is statistically indistinguishable from the same query on the same dataset had any one individual's information been removed. This powerful tool has been adopted by researchers and industry leaders, and has become particularly interesting to machine learning practitioners, who hope to leverage privatized data in training predictive models (Ji et al., 2014; Vietri et al., 2020).

Because differential privacy often depends on adding noise, the results of differentially private algorithms can come at the cost of data accuracy and utility. However, differentially private machine learning algorithms have shown promise across a number of domains. These algorithms can provide tight privacy guarantees while still producing accurate predictions (Abadi et al., 2016). A drawback to most methods, however, is in the one-off nature of training: once the model is produced, the privacy budget for a real dataset can be entirely consumed. The differentially private model is therefore inflexible to retraining and difficult to share/verify: the output model is a black box.

This can be especially disadvantageous in the presence of high dimensional data that require rigorous training techniques like dimensionality reduction or feature selection (Hay et al., 2016). With limited budget to spend, data scientists cannot exercise free range over a dataset, thus sacrificing model quality. In an effort to remedy this, and other challenges faced by traditional differentially private methods for querying, we can use differentially private techniques for synthetic data generation, investigate the privatized data, and train informed supervised learning models.

In order to use the many state-of-the-art methods for differentially private synthetic data effectively in industry domains, we must first address pitfalls in practical analysis, such as the lack of realistic

benchmarking (Arnold & Neunhoeffer, 2020). Benchmarking is non-trivial, as many new state-of-the-art differentially private synthetic data algorithms leverage generative adversarial networks (GANs), making them expensive to evaluate on large scale datasets (Zhao et al., 2019). Furthermore, many of state-of-the-art approaches lack direct comparisons to one another, and by nature of the privatization mechanisms, interpreting experimental results is non-trivial (Jayaraman & Evans, 2019). New metrics presented to analyze differentially private synthetic data methods may themselves need more work to understand, especially in the domain of tabular data (Ruggles et al., 2019; Machanavajjhala et al., 2017).

To that end, our contributions in this paper are 3-fold. (1) We introduce more realisitic benchmarking. Practitioners commonly collect state-of-the-art approaches for comparison in a shared environment (Xu et al., 2019). We provide our evaluation framework, with extensive comparisons on both standard datasets and our real-world, industry applications. (2) We provide experimentation on novel metrics at scale. We stress the tradeoff between synthetic data *utility* and *statistical similarity*, and offer guidelines for untried data. (3) We present a straightforward and pragmatic enhancement, QUAIL, that addresses the tradeoff between *utility* and *statistical similarity*. QUAIL's simple modification to a differentially private data synthesis architecture boosts synthetic data utility in machine learning scenarios without harming summary statistics or privacy guarantees.

## 2 BACKGROUND

Differential Privacy (DP) is a formal definition of privacy offering strong assurances against various re-identification and re-construction attacks (Dwork et al., 2006; 2014). In the last decade, DP has attracted significant attention due to its provable privacy guarantees and ability to quantify privacy loss, as well as unique properties such as robustness to auxiliary information, composability enabling modular design, and group privacy (Dwork et al., 2014; Abadi et al., 2016)

**Definition 1.** (Differential Privacy Dwork et al. (2006)) A randomized function $\mathcal{K}$ provides $(\epsilon, \delta)$-differential privacy if $\forall S \subseteq Range(\mathcal{K})$, all neighboring datasets $D$, $\hat{D}$ differing on a single entry,

$$\Pr[\mathcal{K}(D) \in S] \leq e^{\epsilon} \cdot \Pr[\mathcal{K}(\hat{D}) \in S] + \delta, \tag{1}$$

This is a standard definition of DP, implying that the outputs of differentially private algorithm for datasets that vary by a single individual are indistinguishable, bounded by the privacy parameter $\epsilon$. Here, $\epsilon$ is a non-negative number otherwise known as the *privacy budget*. Smaller $\epsilon$ values more rigorously enforce privacy, but often decrease data utility. An important property of DP is its resistance to post-processing. Given an $(\epsilon, \delta)$-differentially private algorithm $\mathcal{K} : \mathcal{D} \to \mathcal{O}$, and $f : \mathcal{O} \to \mathcal{O}'$ an arbitrary randomized mapping, $f \circ \mathcal{K} : \mathcal{D} \to \mathcal{O}'$ is also differentially private.

Currently, the widespread accessibility of data has increased data protection and privacy regulations, leading to a surge of research into applied scenarios for differential privacy (Allen et al. (2019); Ding et al. (2017); Doudalis et al. (2017). There have been several studies into protecting individual's privacy during model training Li et al. (2014); Zhang et al. (2015); Feldman et al. (2018). In particular, several studies have attempted to solve the problem of preserving privacy in deep learning (Phan et al. (2017); Abadi et al. (2016); Shokri & Shmatikov (2015); Xie et al. (2018); Zhang et al. (2018); Jordon et al. (2018b); Torkzadehmahani et al. (2019)). Here, two main techniques for training models with differential privacy are discussed:

**DP-SGD** Differentially Private Stochastic Gradient Descent (DP-SGD), proposed by Abadi et al. (2016), is one of the first studies to make the Stochastic Gradient Descent (SGD) computation differential private. Intuitively, DPSGD minimizes its loss function while preserving differential privacy by clipping the gradient in the optimization's $l_2$ norm to reduce the model's sensitivity, and adding noise to protect privacy. Further details can be found in the Appendix.

**PATE** Private Aggregation of Teacher Ensembles (PATE) Papernot et al. (2016) provided PATE, which functions by first deploying multiple teacher models that are trained on disjoint datasets, then deploying the teacher models on unseen data to make predictions. On unseen data, the teacher models "vote" to determine the label; here random noise is introduced to privatize the results of the vote. The random noise is generated following the Laplace $Lap(\lambda)$ distribution. PATE further introduces student models, which try to train a model, but only have access to the privatized labels garnered from the teacher's vote. By training multiple teachers on disjoint datasets and adding noise

to the output predicted by those teacher models, the student cannot relearn an individual teacher's model or related parameters.

## 2.1 Privacy Preserving Synthetic Data Models

Synthetic data generation techniques, such as generative adversarial networks (GANs) (Goodfellow et al. (2014); Arjovsky et al. (2017); Xu et al. (2019)), have become a practical way to release realistic fake data for various explorations and analyses. Although these techniques are able to generate high-quality fake data, they may also reveal user sensitive information and are vulnerable to re-identification and/or membership attacks (Hayes et al. (2019); Hitaj et al. (2017); Chen et al. (2019)). Therefore, in the interest of data protection, these techniques must be formally privatized. In recent years, researchers have combined data synthesis methods with DP solutions to allow for the release of data with high utility while preserving an individual's privacy ( Xie et al. (2018); Jordon et al. (2018b); Park et al. (2018); Mukherjee et al. (2019)). Below, we briefly discuss three popular differentially private data synthesizers, evaluated in this paper.

**MWEM** Multiplicative Weights Exponential Mechanism (MWEM) proposed by Hardt et al. (2012) is a simple yet effective technique for releasing differentially private datasets. It combines Multiplicative Weights (Hardt & Rothblum, 2010) with the Exponential Mechanism (McSherry & Talwar, 2007) to achieve differential privacy. The Exponential Mechanism is a popular mechanism for designing $\epsilon$-differentially private algorithms that select for a best set of results $R$ using a scoring function $s(B, r)$. Informally, $s(B, r)$ can be thought of as the quality of a result $r$ for a dataset $B$. MWEM starts with a dataset approximation and uses the Multiplicative Weights update rule to improve the accuracy of the approximating distribution by selecting for informative queries using the Exponential Mechanism. This process of updates iteratively improves the approximation.

**DPGAN** Following Abadi et al. (2016)'s work, a number of studies utilized DP-SGD and GANs to generate differential private synthetic data (Xie et al., 2018; Torkzadehmahani et al., 2019; Xu et al., 2018). These models inject noise to the GAN's discriminator during training to enforce differential privacy. DP's guarantee of post-processing privacy means that privatizing the GAN's discriminator enforces differential privacy on the parameters of the GAN's generator, as the GAN's mapping function between the two functions does not involve any private data. We use the Differentially Private Generative Adversarial Network (DPGAN) Xie et al. (2018) as one of our benchmark synthesizers. DPGAN leverages the Wasserstein GAN proposed by Arjovsky et al. (2017), adds noise on the gradients, and clips the model weights only, ensuring the Lipschitz property of the network. DPGAN has been evaluated on image data and Electronic Health Records (EHR) in the past.

**PATE-GAN** Jordon et al. (2018b) modified the Private Aggregation of Teacher Ensembles (PATE) framework to apply to GANs in order to preserve the differential privacy of synthetic data. Similarly to DPGAN, PATE-GAN only applies the PATE mechanism to the discriminator. The dataset is first partitioned into $k$ subsets, and $k$ teacher discriminators are initialized. Each teacher discriminator is trained to discriminate between a subset of the original data and fake data generated by Generator. The student discriminators are then trained to distinguish real data and fake data using the labels generated by an ensemble of teacher discriminators with random noise added. Lastly, the generator is trained to fool the student discriminator. Jordon et al. (2018b) claim that this method outperforms DPGAN for classification tasks, and present supporting results.

## 3 Enhancing Performance

**The QUAIL Hybrid Method** As we explored generating differentially private synthetic data, we noted a disconnect between the distribution of epsilon, or privacy budget, and the algorithm's application. Generating synthetic data to provide summary statistics necessitates an even distribution of budget across the entire privatization effort; we cannot know a user's query in advance. We may want to reallocate the budget, however, for a known supervised learning task.

QUAIL (Quail-ified Architecture to Improve Learning) is a simple, two model hybrid approach to enhancing the utility of a differentially private synthetic dataset for machine learning tasks. Intuitively, QUAIL assembles a DP supervised learning model in tandem with a DP synthetic data model to produce synthetic data with machine learning potential. *Algorithm 1* describes the procedure more formally.

---

**Algorithm 1: QUAIL pseudocode**

---

**Input:** Dataset $D$, supervised learning target dimension $r'$, budget $\epsilon > 0$, split factor
 $\qquad 0 < p < 1$, size $n$ samples to generate, a differentially private synthesizer $M(D, \epsilon)$, and
 $\qquad$ a differentially private supervised learning model $C(D, \epsilon, t)$ ($t$ is supervisory signal i.e.
 $\qquad$ target dimension). We let $X$ be the universe of samples, and $N$ denote the set of all
 $\qquad$ non-negative integers. Thus, $N^{|X|}$ is all databases in universe $X$, as described in
 $\qquad$ Section 2.3 of Dwork et al. (2014).

**1 Split;**
Split the budget: $\epsilon_M = \epsilon * p$ and $\epsilon_C = \epsilon * (1 - p)$.
Create $D_M$, which is identical to $D$ except $r' \notin D_M$.

**2 In parallel;**

- Train differentially private supervised learning model: $C(D, \epsilon_C, r')$ to produce
  $C_{r'}(s) : N^{|X|} \to R_1$, which can map any arbitrary $s \in N^{|X|}$ to an output label.

- Train differentially private synthesizer: $M(D_M, \epsilon_M) : N^{|X|} \to R_2$ to produce synthesizer
  $M_{D_M}$, which produces synthetic data $S \in N^{|X|}$.

**3 Sample;**

1. Using $M_{D_M}$, generate synthetic dataset $S_{D_M}$ with $n$ samples.

2. For each sample $s_i \in S_{D_M}$, apply $C_{r'}(s_i) = r_i$ i.e. apply model to each synthetic
   datapoint to produce a supervised learning target output $r_i$.

3. Transform $S_{D_M} \to S_R$. For each row $s_i \in S_{D_M}$, $s_i = [s_i, r_i]$ s.t. $\forall s_i, s_i \in dom(D)$ i.e.
   append $r_i$ to each row $s_i$ so that $S_R$ is now in same domain as $D$, the original dataset.

**Output:** Return $S_R$, a synthetic dataset with $n$ samples, where each sample in $S_R$ has target
 $\qquad$ dimension $r_i$ produced by the supervised learner $C_{r'}$

---

**Theorem 3.1** (QUAIL follows the standard composition theorem for $(\epsilon, \delta)$-differential privacy)**.** *The QUAIL method preserves the differential privacy guarantees of $C(R, \epsilon_C, r')$ and $M(R_M, \epsilon_M)$ by the standard composition rules of differential privacy (Dwork et al., 2014).*

*Proof.* Let the first $(\epsilon, \delta)$-differentially private mechanism $M_1 : N^{|X|} \to R_1$ be $C(R, \epsilon_C, r')$. Let the second $(\epsilon, \delta)$-differentially private mechanism $M_2 : N^{|X|} \to R_2$ be $M(R_M, \epsilon_M)$. Fix $0 < p < 1$, $\epsilon_M = p * \epsilon$ and $\epsilon_C = (1 - p) * \epsilon$, then by construction, $\frac{Pr[M_1(x)=(r_1,r_2)]}{Pr[M_2(y)=(r_1,r_2)]} \geq exp(-(\epsilon_M + \epsilon_C))$, which satisfies the differential privacy constraints for a privacy budget of $\epsilon_M + \epsilon_C = \epsilon_{total}$. For more details, see the appendix. $\qquad\square$

**Differentially Private GANs for Tabular Data** In this paper, we focus on *tabular* synthetic data, and explored state-of-the-art methods for generating tabular data with GANs. CTGAN is a state-of-the-art GAN for generating tabular data presented by Xu et al. (2019). We made CTGAN differentially private using the aforementioned techniques, DP-SGD and PATE. CTGAN addresses specific challenges that a vanilla GAN faces when generating tabular data, such as mode-collapse and continuous data following a non-Gaussian distribution (Xu et al., 2019). To model continuous data with multi-model distributions, it leverages mode-specific normalization. In addition, CTGAN introduces a conditional generator, which can generate synthetic rows conditioned by specific discrete columns. CTGAN further trains by sampling, which explores discrete values more evenly.

*DP-CTGAN* Inspired by Xie et al. (2018)'s DPGAN work, we applied DP-SGD to the CTGAN architecture (details can be found in Figure 1 in the Appendix). Similarly to DPGAN, in applying DP-SGD to CTGAN we add random noise to the discriminator and clip the norm to make it differentially private. Based on the post-processing property(Dwork et al., 2014) that any randomized mapping of a differentially private output is also differentially private, the generator is guaranteed to be differentially private when the generator is trained to maximize the probability of $D(G(z))$. In CTGAN, the authors add the cross-entropy loss between conditional vector and produced set of one-hot discrete vectors into the generator loss. To guarantee differential privacy with the generator, we

removed the cross-entropy loss when calculating generator loss. Thus, the generator is differentially private as well. See Figure 1 in Appendix for a diagram.

*PATE-CTGAN* Drawing from work on PATE-GAN, we applied the PATE framework to CTGAN (Jordon et al., 2018b). Similarly to PATE-GAN, we partitioned our original dataset into $k$ subsets and trained $k$ differentially private teacher discriminators to distinguish real and fake data. In order to apply the PATE framework, we further modified CTGAN's teacher discriminator training: instead of using one generator to generate samples, we initialize $k$ conditional generators for each subset of data (shown in Figure 2 in the appendix).

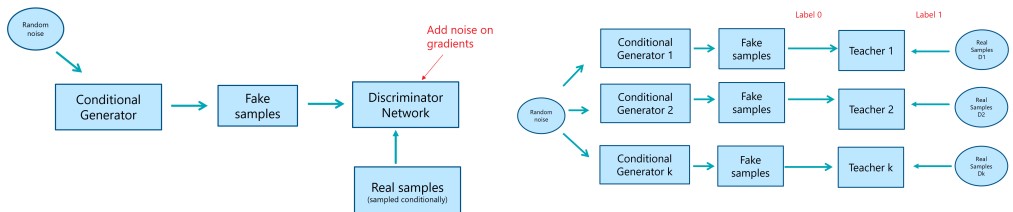

Figure 2: Teacher Discriminator of PATE-
Figure 1: Block diagram of DP-CTGAN model. CTGAN model.

# 4 EVALUATION: METRICS, INFRASTRUCTURE AND PUBLIC BENCHMARKS

We focus on two sets of metrics in our benchmarks: one for comparing the distributional similarity of two datasets and another for comparing the utility of synthetic datasets given a specific predictive task. These two dimensions should be viewed as complementary, and in tandem they capture the overall quality of the synthetic data.

*Distributional similarity* To provide a quantitative measure for comparison of synthetically generated datasets, we use a relatively new metric for assessing synthetic data quality: *propensity score mean-squared error (pMSE) ratio score*. Proposed by Snoke & Slavković (2018), pMSE provides a statistic to capture the distributional similarity between two datasets. Given two datasets, we combine the two together with an indicator to label which set a specific observation comes from. A discriminator is then trained to predict these indicator labels. To calculate pMSE, we simply compute the mean-squared error of the predicted probabilities for this classification task. If our model is unable to discern between these classes, then the two datasets are said to have high distributional similarity. To help limit the sensitivity of this metric to outliers, Snoke & Slavković (2018) propose transforming pMSE to a ratio by leveraging an approximation to the null distribution. For the ratio, we simply divide the pMSE by the expectation of the null distribution. A ratio score of 0 implies the two datasets are identical.

*Machine Learning Utility* Given the context of this paper, we aim to provide quantitative measures for approximating the utility of differentially private synthetic data in regards to machine learning tasks. Specifically, we used three metrics: *AUCROC* and *F1-score*, two traditional utility measures, and the *synthetic ranking agreement (SRA)*, a more recent measure. SRA can be thought of as the probability that a comparison between any two algorithms on the synthetic data will be similar to comparisons of the same two algorithms on the real data (Jordon et al., 2018a). Descriptions of each metric can be found in the Appendix.

**Evaluation Infrastructure** The design of our pipeline addressed scalability concerns, allowing us to benchmark four computationally expensive GANs on five high dimensional datasets across the privacy budgets $\epsilon = [0.01, 0.1, 0.5, 1.0, 3.0, 6.0, 9.0]$, averaged across 12 runs. We used varying compute, including CPU nodes (24 Cores, 224 GB RAM, 1440 GB Disk) and GPU nodes $GPU$ (4 x NVIDIA Tesla K80). Despite extensive computational resources, we could not adequately address the problem of hyperparameter tuning differentially private algorithms for machine learning tasks, which is an open research problem (Liu & Talwar, 2019). In our case, a grid search was computationally intractable: for each run of the public datasets on all synthesizers, Car averaged 1.27 hours, Mushroom averaged 8.33 hours, Bank averaged 13.30 hours, Adult averaged 14.47 hours and Shopping averaged 27.37 hours. We trained our GANs using the experimentally determined

hyperparameters, and were informed by prior work around each algorithm. We include a description of the parameters used for each synthesizer in the appendix.

*Regarding F1-score and AUC-ROC*: We averaged across the maximum performance of five classification models: an AdaBoost classifier, a Bagging classifier, a Logistic Regression classifier, Multi-layer Perceptron classifier, and a Random Forest classifier. We decided to focus on one classification scenario specifically: train-synthetic test-real or TSTR, which was far more representative of applied scenarios than train-synthetic test-synthetic. We compare these values to train-real test-real (TRTR).

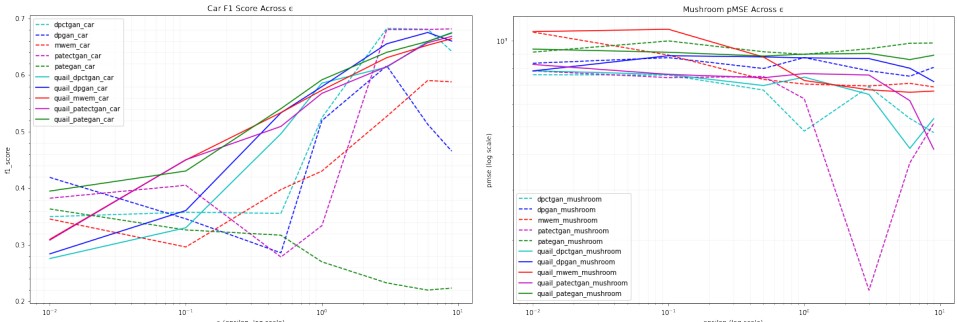

Figure 3: Real Car F1 Score: 0.97     Figure 4: Mushroom pMSE

**Experimental Results: Public Datasets** We ran experiments on five public real datasets, which helped inform the applied scenario discussed in Section 5. Full details of the experiments can be found in the Appendix, Figures 25-30. We will refer to the datasets as Adult, Car, Mushroom, Bank and Shopping, and will discuss a handful of the results here in terms of their machine learning *utility* and their *statistical similarity* to the real data. The individual synthesizer's are color coded consistently across plots, and their performance is tracked according to dataset (so "dpctgan_car" tracks the graphed metric for DPCTGAN on the Car dataset).

In our Car evaluations in Figure 3, we see strong performance from the QUAIL variants on very low $\epsilon$ values. However, we note that for $\epsilon \geq 3.0$, DPCTGAN and PATECTGAN outperform even the QUAIL enhanced models. We further note that PATECTGAN performs remarkably well on the pMSE metric across $\epsilon$ values in Figure 28b. In our Mushroom evaluations in Figure 28a, QUAIL variants also outperformed other synthesizers. However, PATECTGAN's exhibits the best *statistical similarity* (pMSE score) with larger $\epsilon$. In our evaluations on the Adult dataset in Figure 30a, while PATECTGAN performs well, DPCTGAN performs best when $\epsilon \geq 3.0$.

Our findings suggest that generally, with larger budgets ($\epsilon \geq 3.0$), PATECTGAN improves on other synthesizers, both in terms of *utility* and *statistical similarity*. With smaller budgets ($\epsilon \leq 1.0$), DPCTGAN may perform better. Synthesizers are not able to achieve reasonable *utility* under low budgets ($\epsilon \leq 0.1$), but DPCTGAN was able to achieve *statistical similarity* in this setting.

## 5   EVALUATION: APPLIED SCENARIO

Supported by learnings from experiments on the public datasets, we evaluated our benchmark DP synthesizers on several private internal datasets, for different scenarios such as classification and regression. We show that DP synthetic data models can perform on real-world data, despite a noisy supervised learning problem and skewed distributions when compared to the more standardized public datasets.

**Classification** The data used in this set of experiment include ~100,000 samples and 30 features. The data includes only categorical columns each containing between 2 to 24 categories. One of our tasks with this dataset was to train a classification task with three classes. We faced significant challenges when managing the long-tail distribution of each feature. Figure 26, which can be found in the appendix, shows an example of data distributions for different attributes in this data.

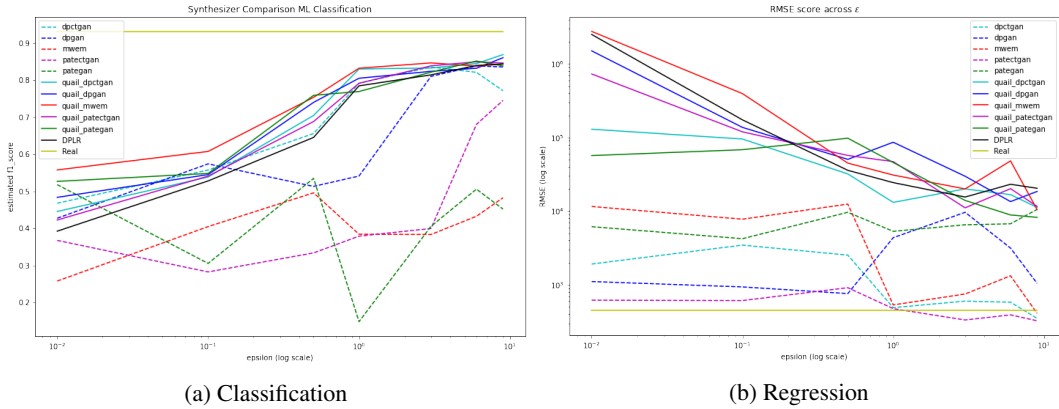

(a) Classification         (b) Regression

Figure 5: ML evaluation results for internal dataset

We ran our evaluation suite on the applied internal data scenarios to generate the synthetic data from each DP synthesizer and benchmark standard ML models. We also applied a Logistic Regression classifier with differential privacy from IBM. (Chaudhuri et al., 2011; diffprivlib) to the real data as a baseline. Figure 5a shows the ML results from our evaluation suite. As expected, as the privacy budget $\epsilon$ increases, performance generally improves. DP-CTGAN had the highest performance without the QUAIL enhancement. QUAIL, however, improved the performance of all synthesizers. In particular, a QUAIL enhanced DPCTGAN synthesizer had the highest performance across epsilons in this experiment. In particular, these experiments demonstrated the advantages of QUAIL, combining DP synthesizers with a DP classifier for a classification model.

**Regression** In this experiment, we used another internal data for the task of regression. Our dataset included 27466 and 6867 training and testing samples, respectively. The domain comprised eight categorical and 40 continuous features. After generating the DP synthetic data from each model, we used Linear Regression to predict the target variable. Figure 5b shows the results from the evaluation suite. We used RMSE as the evaluation metric. For QUAIL boosting, we used a Linear Regression model with differential privacy from IBM (Sheffet, 2015; diffprivlib). We also compared the DP synthesizers with a "vanilla" DP Linear Regression (DPLR) using real data.

In this experiment, PATECTGAN outperformed other models and even improved on the RMSE (root-mean-squared-error) when compared to the real data for budget $\epsilon > 1.0$. For QUAIL-enhanced models, the RMSE is considerably larger than the real and other DP synthetic data. We attribute this to a weakness of the embedded regression model (DP Linear Regression) in QUAIL for this data scenario. Based on our observations, small privacy budgets ($\epsilon < 10.0$) for DP Linear Regression significantly affects its performance. However, as shown in Figure 5b, we still see some boost on the QUAIL variant synthesizers when compared to the "vanilla" DP Linear Regression. For distributional similarity comparison, please refer to Figure 27 in the appendix.

**QUAIL Evaluations** QUAIL's hyperparameter, the split factor $p$ where $0 < p < 1$, determines the distribution of budget between classifier and synthesizer. We generated classification task datasets with 10000-50000 samples, 7 feature columns and 10 output classes using the $make\_classification$ package from Scikit-learn (Pedregosa et al., 2011). We experimented with the values $p = [0.1, 0.3, 0.5, 0.7, 0.9]$, and report on results, varying budget $\epsilon = [1.0, 3.0, 10.0]$. See the appendix for complete results and a list of DP classifiers we experimented on embedding in QUAIL.

Our figures represent the delta $\delta$ in F1 score between training the classifier $C(R, \epsilon_C, r')$ on the original dataset (the "vanilla" scenario) ($F1_v$), and training a Random Forest classifier on the differentially private synthetic dataset produced by applying QUAIL to an hybrid of $C(R, \epsilon_C, r')$ and one of our benchmark synthesizers $M(D, \epsilon_M)$ ($F1_q$). We plot $\delta = F1_v - F1_q$ across epsilon splits and datasizes. Positive to highly positive deltas are grey→red, indicating the "vanilla" scenario outperformed the QUAIL scenario. Small or negative deltas are blue, indicating the QUAIL scenario matched, or even outperformed, the "vanilla" scenario. Each cell contains $\delta$ for some $p$ on datasets $|10000 - 50000|$. In our results we use DP Gaussian Naive Bayes (DP-GNB) as $C(R, \epsilon_C, r')$ ($F1_v$),

and trained a Random Forest Classifier on data generated by QUAIL (F1$_q$) (recall QUAIL combines $C(R, \epsilon_C, r')$ and a DP synthesizer) (Vaidya et al., 2013; diffprivlib). We average across 75 runs.

Note the correlation between epsilon split, datasize and classification performance when embedding PATECTGAN in QUAIL, shown in Figure 6, suggesting that a higher $p$ split value increases the likelihood of outperforming $C(R, \epsilon_C, r')$. For an embedded MWEM synthesizer, seen in Figure 7, the relationship between split, scale and performance was more ambiguous. In general, a higher split factor $p$, which assigns more budget to the differentially private classifier $C(D, \epsilon_M, t)$ could improve the utility of the overall synthetic dataset. However, any perceived improvements were highly dependant on the differentially private synthesizer used. Our QUAIL results are agnostic to the embedded supervised learning algorithm $C(R, \epsilon_C, r')$, as they depict relative performance, though different methods of supervised learning are more suitable to certain domains. Future work might explore alternative classifiers or regression models, and how purposefully *overfitting* the model $C(R, \epsilon_C, r')$ could contribute to improved synthetic data.

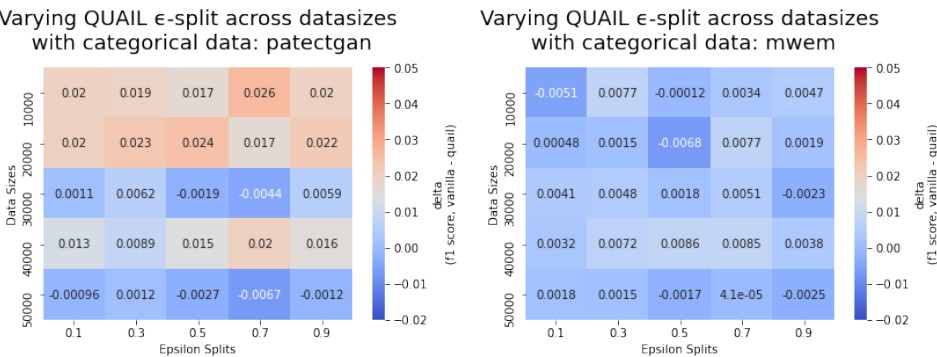

Figure 6: Privacy budget $\epsilon = 3.0$          Figure 7: Privacy budget $\epsilon = 3.0$

**Peeling Back QUAIL: Analysis of via clustering and direct comparison** By first assessing the TSNE clustering in Figures 8 and 9, we see that not only is the synthetic data produced by QUAIL very similar to the real data, but the accuracy of the labeling for the embedded model (in this case, DPLR) is also very similar. Further investigation into data scale revealed that the QUAIL method takes advantage of allocating excess epsilon when datasets are large. As datascale increases, the sensitivity of the differentially private model decreases and so less epsilon can be used more efficiently. Thus, we see that an exaggerated difference between DPLR embedded in QUAIL (with an epsilon of 2.4) and DPLR with an epsilon of 3.0 for a dataset of 20,000 samples. In this case, the embedded DPLR model accuracy suffers, and so does the learning utility of the produced synthetic data. Conversely, as we increase the data size to 50,000 and 100,000 samples, we see that the internal model (with epsilon 2.4) can match the performance of the vanilla model (with epsilon 3.0). Then, the synthetic dataset serves only to augment the performance by small but significant margin (in Figure 10, we see a bump of three percent to f1 score.)

**Time Performance Analysis of QUAIL: Making supervised learning more efficient** QUAIL benefits the efficiency of training intensive GANs. In Table 1, the time performance of QUAIL is compared with non-Quail methods. Specifically, we select two epsilons ($\epsilon = 3.0$ and $\epsilon = 6.0$) and two QUAIL split factors ($p = 0.9$ and $p = 0.5$). From this table, it can be seen that in all GAN-based models, QUAIL can improve time efficiency considerably. This is more noticeable as the epsilon increases where training time for models such as DPCTGAN and DPGAN skyrockets.

## 6 PUNCHLINES

We summarize our findings in the following *punchlines*, concise takeaways from our work for researchers and applied practitioners exploring DP synthetic data.

1. Holistic Performance. *No single model performed best always (but PATECTGAN performed well often).* Model performance was domain dependent, with continuous/categorical features, dataset scale and distributional complexity all affecting benchmark results. However, in general, we found

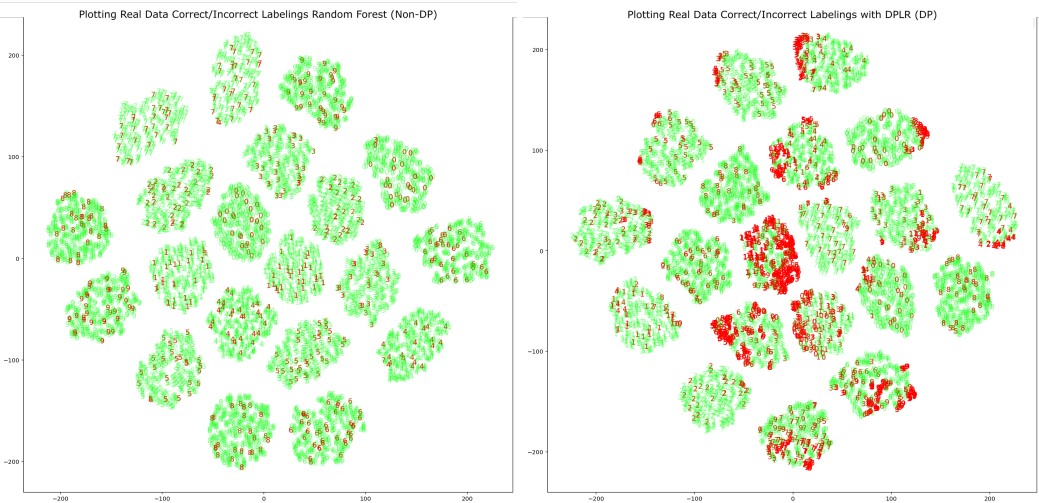

Figure 8: Privacy budget $\epsilon = 3.0$

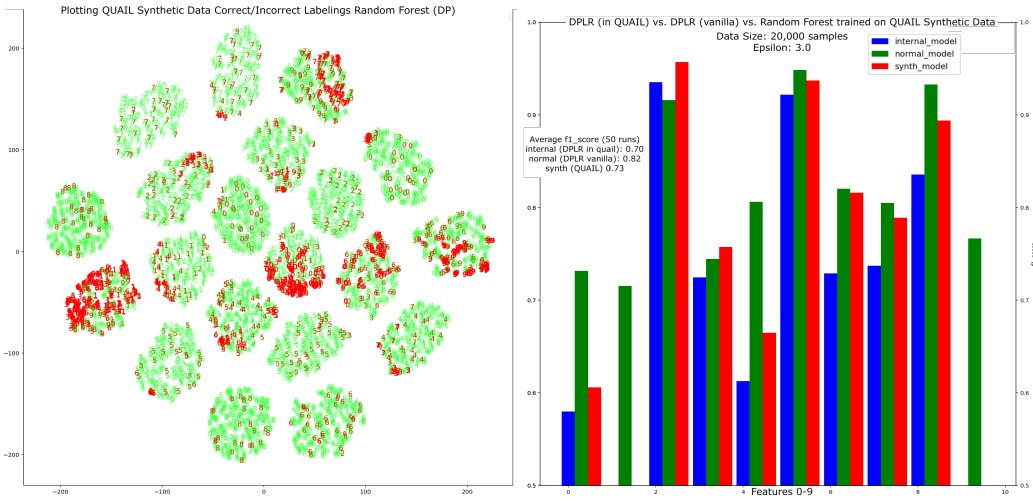

Figure 9: Privacy budget $\epsilon = 3.0$

that PATECTGAN had better *utility* and *statistical similarity* in scenarios with high privacy budget ($\epsilon >= 3.0$) when compared to the other synthesizers we benchmarked. Conversely, with low privacy budget ($\epsilon <= 1.0$) we found that DPCTGAN had better *utility*, but PATECTGAN may still be better in terms of *statistical similarity*.

2. Computational tradeoff. *Our highest performant GANs were slow, and MWEM is fast.* PATECT-GAN and DPCTGAN, while being our most performant synthesizers, were also the slowest to train. With GANs, more computation often correlates with higher performance (Lucic et al., 2018). On categorical data, MWEM performed competitively, and is significantly faster to train in any domain.

3. Using AUC-ROC and F1 Score. *One should calculate both, especially to best understand QUAIL's tradeoffs.* Our highest performing models by F1 Score often had QUAIL enhancements, which sometimes, but not always, detrimentally affected AUC-ROC. Without both metrics, one risks using a QUAIL enhancement for a model with high training accuracy that struggles to generalize.

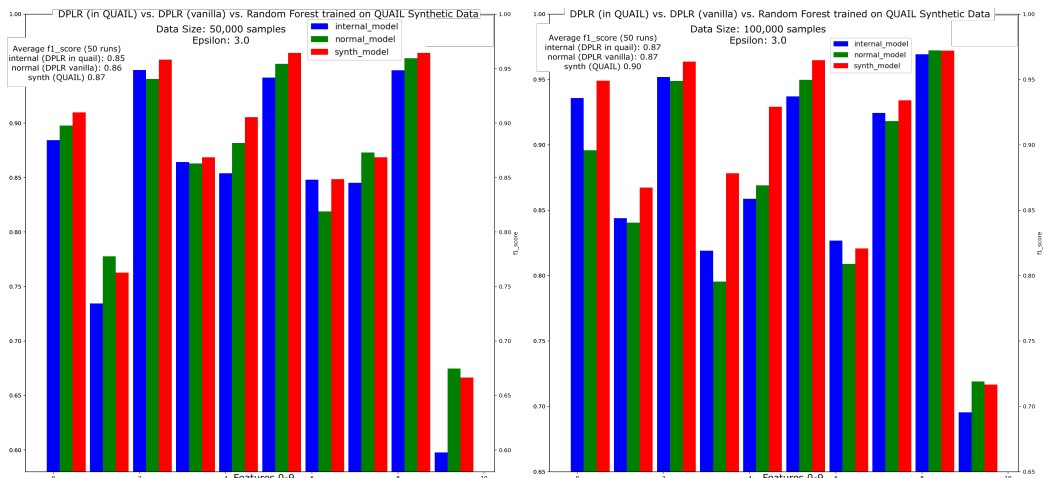

Figure 10: Privacy budget $\epsilon = 3.0$

| method | classification | | regression | |
|---|---|---|---|---|
| | $\epsilon$: 3 | $\epsilon$: 6 | $\epsilon$: 3 | $\epsilon$: 6 |
| dpgan | 1130 | 3950 | 1300 | 4335 |
| quail_dpgan_p0.9 | 20 | 58 | 202 | 203 |
| quail_dpgan_p0.5 | 305 | | 444 | 1323 |
| dpctgan | 15689 | 24808 | 3989 | 14029 |
| quail_dpctgan_p0.9 | 199 | 759 | 198 | 318 |
| quail_dpctgan_p0.5 | 4478 | | 1171 | 3964 |
| pategan | 77 | 206 | 224 | 389 |
| quail_pategan_p0.9 | 18 | 15 | 152 | 156 |
| quail_pategan_p0.5 | 31 | 69 | 165 | 210 |
| patectgan | 160 | 449 | 263 | 474 |
| quail_patectgan_p0.9 | 16 | 19 | 152 | 155 |
| quail_patectgan_p0.5 | 56 | 131 | 185 | 270 |

Table 1: Time Performance Analysis of QUAIL compared to other synthesizers (time is shown in seconds)

4. Using pMSE. *pMSE can be used alongside ML utility metrics to balanced experiments.* pMSE concisely captures *statistical similarity*, and allows practitioners to easily balance *utility* against the distributional quality of their synthetic data.

5. Enhancing with QUAIL. *QUAIL's effectiveness depends far more on the quality of the embedded differentially private classifier than on the synthesizer.* QUAIL showed promising results in almost all the scenarios we evaluated. Given confidence in the embedded "vanilla" differentially private classifier, QUAIL can be used regularly to improve the utility of DP synthetic data.

6. Reservations for use in applied scenarios. *Applied DP for ML is hard, thanks to scale and dimensionality.* Applied scenarios we presented assessed large datasets, leading to high computational costs that makes tuning performance difficult. Dimensionality is tricky to deal with in large, sparse, imbalanced private applied scenarios (like we faced with internal datasets). Practitioners may want to investigate differentially private feature selection or dimensionality reductions before training. We are aware of work being done to embed autoencoders into differentially private synthesizers, and view this a promising approach (Nguyen et al., 2020).

## 7 CONCLUSION

With this paper, we set out to assess the efficacy of differentially private synthetic data for use on machine learning tasks. We surveyed an histogram based approach (MWEM) and four differentially private GANs for data synthesis (DPGAN, PATE-GAN, DPCTGAN and PATECTGAN). We evaluated each approach using an extensive benchmarking pipeline. We proposed and evaluated QUAIL, a straightforward method to enhance synthetic data *utility* in ML tasks. We reported on results from two applied internal machine learning scenarios. Our experiments favored PATECTGAN when the privacy budget $\epsilon \geq 3.0$, and DPCTGAN when the privacy budget $\epsilon \leq 1.0$. We discussed nuances of domain-based tradeoffs and offered takeaways across current methods of model selection, training and benchmarking. As of writing, our experiments represent one of the largest efforts at benchmarking differentially private synthetic data, and demonstrates the promise of this approach when tackling private real-world machine learning problems.

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

## A    APPENDIX

## B    METHODS

### B.1    DP-SGD DETAILED STEPS

The detailed training steps are as follows:

1. A batch of random samples is taken and the gradient for each sample is computed

2. For each computed gradient $g$, it is clipped to $g/max(1, \frac{\|g\|_2}{C})$, where $C$ is a clipping bound hyperparameter.

3. A Gaussian noise ($\mathcal{N}(0, \sigma^2 C^2 I)$) (where $\sigma$ is the noise scale) is added to the clipped gradients and the model parameters are updated.

4. Finally, the overall privacy cost ($\epsilon, \delta$) is computed using a privacy accountant method.

### B.2    DESCRIPTIONS OF METRICS: F1-SCORE, AUC-ROC AND SRA

*F1-score* measures the accuracy of a classifier, essentially calculating the mean between precision and recall and favoring the lower of the two. It varies between 0 and 1, where 1 is perfect performance.

*AUC-ROC*: Area Under the Receiver Operating Characteristic (AUC-ROC) represents the Receiver Operating Characteristic curve in a single number between 0 and 1. This provides insight into the true positive vs. false positive rate of the classifier.

*SRA*: SRA can be thought of as the probability that a comparison between any two algorithms on the synthetic data will be similar to comparisons of the same two algorithms on the real data. SRA compares train-synthetic test-real (i.e. TSTR, which uses differentially private synthetic data to train the classifier, and real data to test) with train-real test-real (TRTR, which uses differentially private synthetic data to train and test the classifier)

*Further Motivation* Machine learning practitioners often need a deep understanding of data in order to train predictive models. That can be incredibly difficult when data is private. Training one-off, blackbox "vanilla" DP classifiers cannot be retrained, as this risks individual privacy, making parameter tuning and feature selection incredibly difficult with these models. Differentially private synthetic data allows practitioners to treat data normally, without further privacy considerations, giving them an opportunity to fine tune their models.

## C    QUAIL

### C.1    QUAIL FURTHER DETAILS

We evaluated with a few vanilla differentially private classifiers $C(R, \epsilon_C, r')$:

1. Logistic Regression classifier with differential privacy. (Chaudhuri et al., 2011; diffprivlib)

2. Gaussian Naive Bayes with differential privacy. (Vaidya et al., 2013; diffprivlib)

3. Multi-layer Perceptron (Neural Network) with differential privacy. (Abadi et al., 2016)

**Theorem C.1.** *Standard Composition Theorem (Dwork et al., 2014) Let $M_1 : N^{|X|} \to R_1$ be an $\epsilon_1$-differentially private algorithm, and let $M_2 : N^{|X|} \to R_1$ be $\epsilon_2$-differentially private algorithm. Then their combination, defined to be $M_{1,2} \to R_1 X R_2$ by the mapping: $M_{1,2}(x) = (M_1(x), M_2(x))$ is $\epsilon_1 + \epsilon_2$-differentially private.*

*Proof.* Let $x, y \in N^{|X|}$ be such that $||x - y||_1 < 1$. Fix any $(r_1, r_2) \in R_1 X R_2$. Then:

$$\frac{Pr[M_{1,2}(x) = (r_1, r_2)]}{Pr[M_{1,2}(y) = (r_1, r_2)]} = \frac{Pr[M_1(x) = r_1]Pr[M_2(x) = r_2]}{Pr[M_1(y) = r_1]Pr[M_2(y) = r_2]} \tag{2}$$

$$= (\frac{Pr[M_1(x) = r_1]}{Pr[M_1(y) = r_1]})(\frac{Pr[M_2(x) = r_2]}{Pr[M_2(y) = r_2]}) \tag{3}$$

$$\leq exp(\epsilon_1)exp(\epsilon_2) \tag{4}$$

$$= exp(\epsilon_1 + \epsilon_2) \tag{5}$$

By symmetry, $\frac{Pr[M_{1,2}(x)=(r_1,r_2)]}{Pr[M_{1,2}(y)=(r_1,r_2)]} \geq exp(-(\epsilon_1 + \epsilon_2))$ $\qquad\square$

*Proof. QUAIL: full proof of differential privacy* Let the first $(\epsilon, \delta)$-differentially private mechanism $M_1 : N^{|X|} \to R_1$ be $C(R, \epsilon_C, r')$. Let the second $(\epsilon, \delta)$-differentially private mechanism $M_2 : N^{|X|} \to R_2$ be $M(R_M, \epsilon_M)$. Fix $0 < p < 1$, then by construction, with original budget $B = \epsilon$,

$$B = \epsilon = p * \epsilon + (1 - p) * \epsilon \tag{6}$$

$$\epsilon_M = p * \epsilon \tag{7}$$

$$\epsilon_C = (1 - p) * \epsilon \tag{8}$$

$$\text{By the } Standard\ Composition\ Theorem \tag{9}$$

$$\frac{Pr[M_{C,M}(x) = (r_1, r_2)]}{Pr[M_{C,M}(y) = (r_1, r_2)]} \geq exp(-(\epsilon_M + \epsilon_C)) \tag{10}$$

$$\frac{Pr[M_{C,M}(x) = (r_1, r_2)]}{Pr[M_{C,M}(y) = (r_1, r_2)]} \geq exp(-(B)) \tag{11}$$

$\qquad\square$

## C.2 QUAIL FULL RESULTS



Figure 11: Budget $\epsilon = 1.0$  Figure 12: Budget $\epsilon = 3.0$  Figure 13: Budget $\epsilon = 10.0$



Figure 14: Budget $\epsilon = 1.0$  Figure 15: Budget $\epsilon = 3.0$  Figure 16: Budget $\epsilon = 10.0$

## D EVALUATIONS

### D.1 DESCRIPTION OF INFRASTRUCTURE

Our experimental pipeline provides an extensible interface for loading datasets from remote hosts, specifically from the UCI ML Dataset repository (Dua & Graff, 2017). For each evaluation dataset,

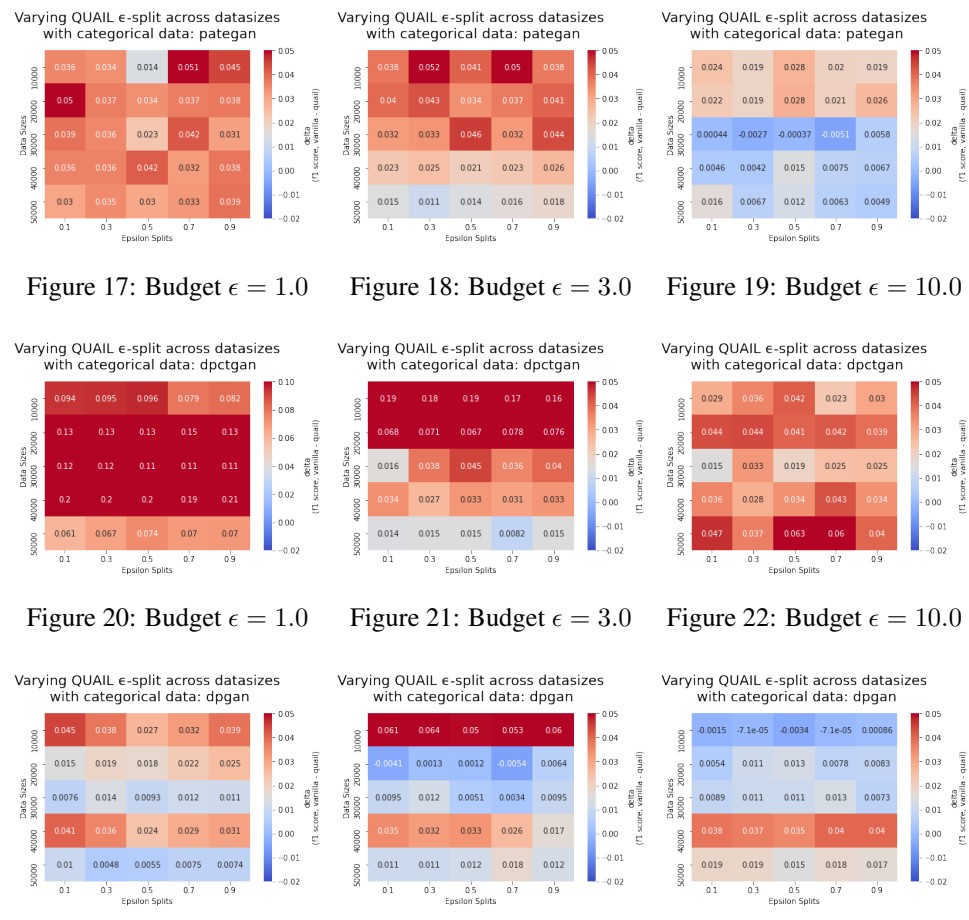

Figure 17: Budget $\epsilon = 1.0$     Figure 18: Budget $\epsilon = 3.0$     Figure 19: Budget $\epsilon = 10.0$

Figure 20: Budget $\epsilon = 1.0$     Figure 21: Budget $\epsilon = 3.0$     Figure 22: Budget $\epsilon = 10.0$

Figure 23: Budget $\epsilon = 1.0$     Figure 24: Budget $\epsilon = 3.0$     Figure 25: Budget $\epsilon = 10.0$

we launch a process that synthesizes datasets for each privacy budget ($\epsilon$s) specified on each synthesizer specified. Once the synthesis is complete, the pipeline launches a secondary process that analyzes the synthetic data, training classifiers and running the previously mentioned novel metrics. The run is launched, and the results are logged, using MLFlow runs (Zaharia et al., 2018) with an Azure Machine Learning compute-cluster backend. Our compute used CPU nodes $STANDARD\_NC24r$ (24 Cores, 224 GB RAM, 1440 GB Disk) and GPU nodes $GPU$ (4 x NVIDIA Tesla K80). We highly encourage future work into hyperparameter tuning for differentially private machine learning tasks, and believe our evaluation pipeline could be of some use in that effort.

## D.2 DETAILS ON DATA

Results presented on the Public Datasets are averaged across 12 runs. SRA results were moved to the appendix after difficulty interpreting their significance, although there are potential trends that warrant further exploration.

Below is a list of parameters used in training:

```
DPCTGAN
    embedding_dim=128,
    gen_dim=(256, 256),
    dis_dim=(256, 256),
    l2scale=1e-6,
    batch_size=500,
    epochs=300,
```

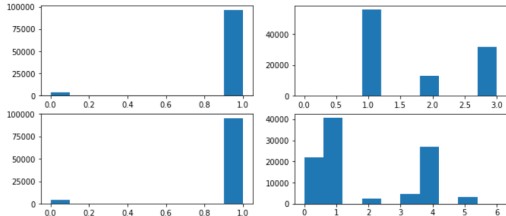

Figure 26: Data distribution of the internal dataset for various attributes. Included to highlight the imbalanced nature, difficulty of supervised learning problem.

| Dataset Name | Samples | Continuous Features | Categorical Features | Total Features | Class Distributions | UCI Link |
|---|---|---|---|---|---|---|
| Adult | 48842 | 6 | 8 | 14 | 24.78% positive (binary imbalanced) | UCI |
| Bank Marketing | 45211 | 8 | 12 | 20 | N/A (binary) | UCI |
| Car Evaluation | 1728 | 0 | 6 | 6 | 0 - 70.023 %, 1 - 22.222 %, 2 - 3.993 %, 3 - 3.762 % (multiclass imbalanced) | UCI |
| Mushroom | 8124 | 0 | 22 | 22 | 51.8% positive (binary balanced) | UCI |
| Online Shoppers Purchasing Intention (Shopping) | 12330 | 10 | 8 | 18 | 84.5% negative (binary imbalanced) | UCI |

Table 2: Details on Public Datasets used for benchmarking.

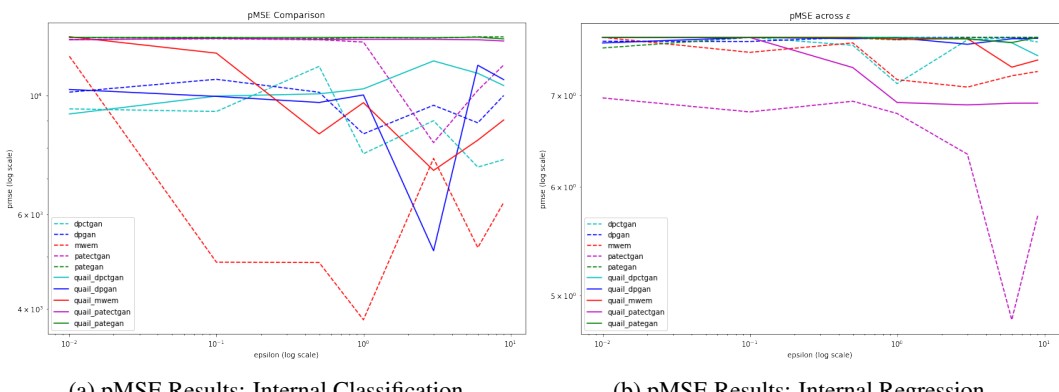

(a) pMSE Results: Internal Classification      (b) pMSE Results: Internal Regression

Figure 27: pMSE evaluation results for internal dataset. PATECTGAN performed best in both cases.

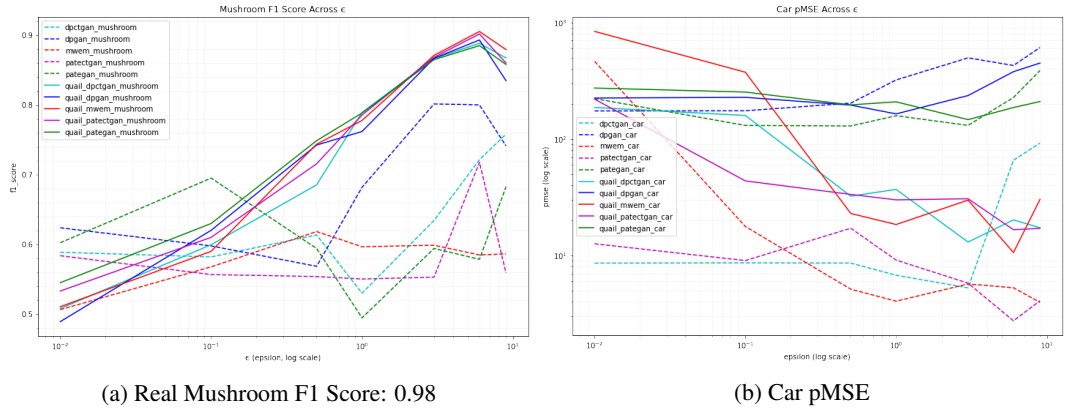

(a) Real Mushroom F1 Score: 0.98

(b) Car pMSE

Figure 28: PATECTGAN demonstrated better performance at higher epsilons. QUAIL synthesizers performed best at low epsilon privacy values.

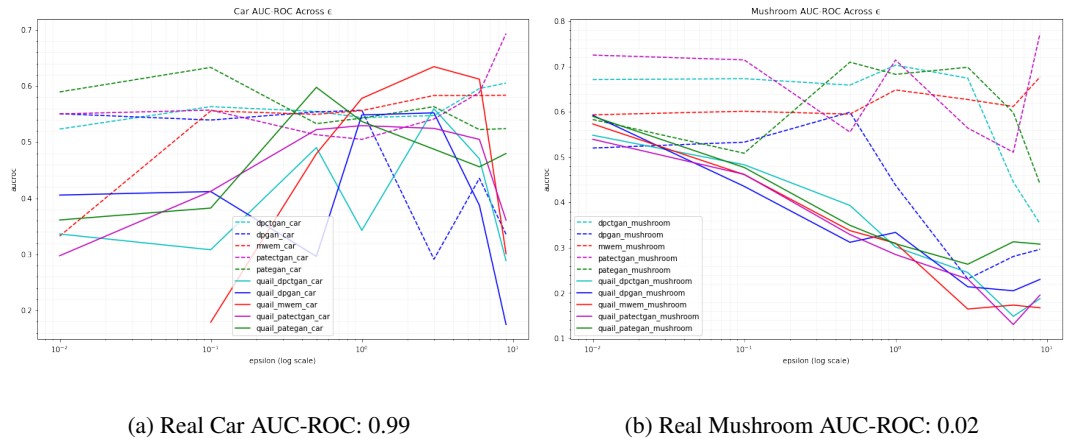

(a) Real Car AUC-ROC: 0.99

(b) Real Mushroom AUC-ROC: 0.02

Figure 29: Mushrooms AUC-ROC curve demonstrated that the QUAIL synthesizers might not generalize particularly well.

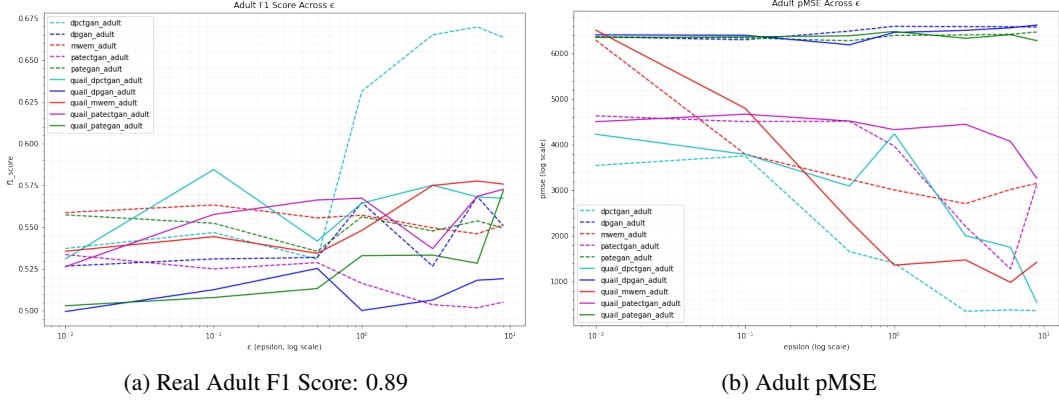

(a) Real Adult F1 Score: 0.89

(b) Adult pMSE

Figure 30: DPCTGAN outperformed other synthesizers by significant margins with Adult, both in terms of ML utility and statistical similarity.

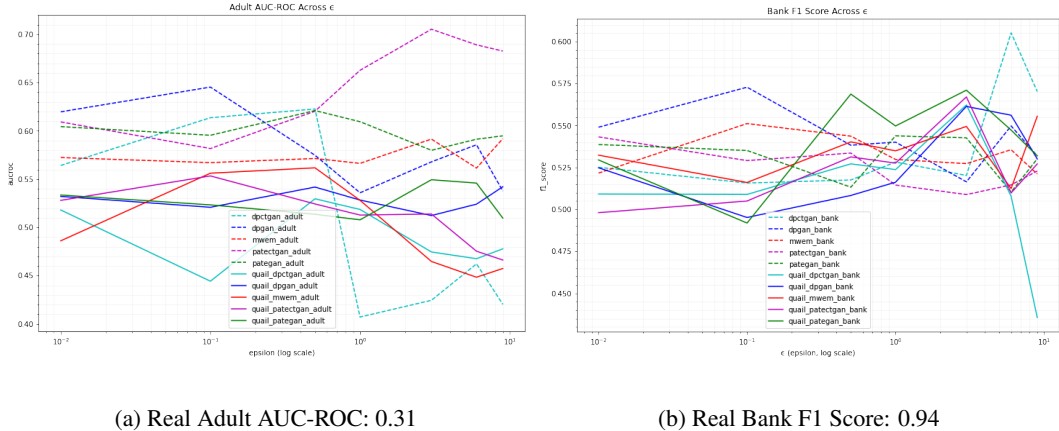

(a) Real Adult AUC-ROC: 0.31

(b) Real Bank F1 Score: 0.94

Figure 31: As the most complex benchmark dataset, Bank presented a particular challenge. The results are difficult to interpret, and would require further experimentation to draw conclusions.

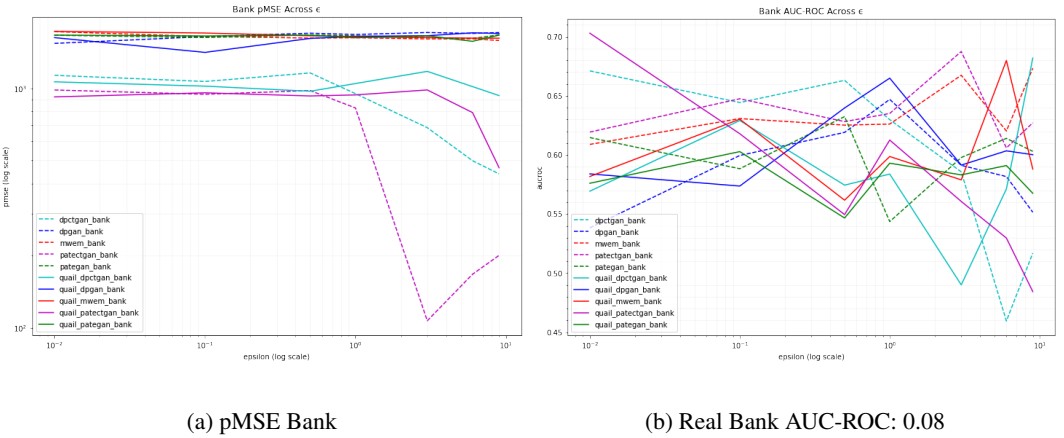

(a) pMSE Bank

(b) Real Bank AUC-ROC: 0.08

Figure 32: Despite the noisy plots, at higher epsilon values, based F1-Scores and this pMSE plot, it does appear as though DPCTGAN and PATECTGAN improved on the other synthesizers.

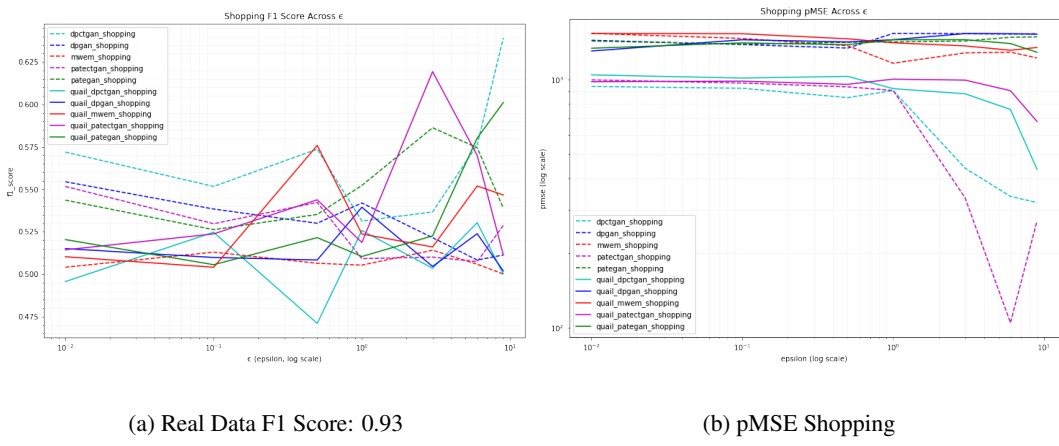

(a) Real Data F1 Score: 0.93

(b) pMSE Shopping

Figure 33: We see similar results here to our Bank dataset. Bank and Shopping appear to have been too complex for the synthesizers at the relatively low epsilon budgets provided.

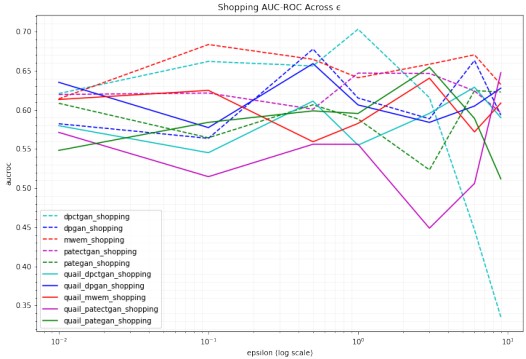

(a) Real Data AUC-ROC: 0.09

(a) SRA results for MWEM

| epsilons | bank | car | shopping | mushroom | adult |
|---|---|---|---|---|---|
| 0.01 | 0.4 | 0.1 | 0.3 | 0.9 | 0.3 |
| 0.10 | 0.4 | 0.3 | 0.6 | 0.9 | 0.5 |
| 0.50 | 0.7 | 0.2 | 0.6 | 0.7 | 0.6 |
| 1.00 | 0.7 | 0.1 | 0.6 | 0.8 | 0.8 |
| 3.00 | 0.3 | 0.3 | 0.5 | 0.9 | 0.2 |
| 6.00 | 0.5 | 0.3 | 0.6 | 0.9 | 0.2 |
| 9.00 | 0.6 | 0.3 | 0.6 | 0.7 | 0.4 |

(b) SRA results for PATEGAN

| epsilons | bank | car | shopping | mushroom | adult |
|---|---|---|---|---|---|
| 0.01 | 0.8 | 0.5 | 0.8 | 0.6 | 0.4 |
| 0.10 | 0.7 | 0.4 | 0.7 | 0.3 | 0.8 |
| 0.50 | 0.5 | 0.7 | 0.4 | 0.9 | 0.5 |
| 1.00 | 0.6 | 0.4 | 0.8 | 0.9 | 0.8 |
| 3.00 | 0.8 | 0.7 | 0.7 | 1.0 | 0.7 |
| 6.00 | 0.6 | 0.9 | 0.6 | 0.3 | 0.8 |
| 9.00 | 0.5 | 0.6 | 0.5 | 0.7 | 0.6 |

```
pack=1,
log_frequency=True,
disabled_dp=False,
target_delta=None,
sigma = 5,
max_per_sample_grad_norm=1.0,
verbose=True,
loss = 'wasserstein'
```

PATECTGAN
```
embedding_dim=128,
gen_dim=(256, 256),
dis_dim=(256, 256),
l2scale=1e-6,
epochs=300,
pack=1,
```

(a) SRA results for DPGAN

| epsilons | bank | car | shopping | mushroom | adult |
|---|---|---|---|---|---|
| 0.01 | 0.6 | 0.2 | 0.7 | 0.7 | 0.4 |
| 0.10 | 0.4 | 0.1 | 0.7 | 0.7 | 0.7 |
| 0.50 | 0.4 | 0.4 | 0.9 | 0.6 | 0.3 |
| 1.00 | 0.8 | 0.2 | 0.7 | 1.0 | 0.5 |
| 3.00 | 0.9 | 0.4 | 0.2 | 1.0 | 0.9 |
| 6.00 | 0.9 | 0.6 | 0.5 | 0.6 | 0.8 |
| 9.00 | 0.4 | 0.2 | 0.9 | 0.9 | 0.7 |

(b) SRA results for PATECTGAN

| epsilons | bank | car | shopping | mushroom | adult |
|---|---|---|---|---|---|
| 0.01 | 0.4 | 0.4 | 0.7 | 1.0 | 0.5 |
| 0.10 | 0.7 | 0.3 | 0.5 | 0.9 | 0.5 |
| 0.50 | 0.5 | 0.8 | 0.7 | 0.9 | 0.4 |
| 1.00 | 0.5 | 0.3 | 0.8 | 0.6 | 0.5 |
| 3.00 | 0.7 | 0.4 | 0.5 | 0.8 | 0.3 |
| 6.00 | 0.5 | 0.4 | 0.5 | 1.0 | 0.4 |
| 9.00 | 0.4 | 0.3 | 0.5 | 0.8 | 0.5 |

(a) SRA results for DPCTGAN

| epsilons | bank | car | shopping | mushroom | adult |
|---|---|---|---|---|---|
| 0.01 | 0.5 | 0.1 | 0.8 | 0.9 | 0.0 |
| 0.10 | 0.3 | 0.1 | 0.6 | 0.9 | 0.2 |
| 0.50 | 0.9 | 0.2 | 0.5 | 0.7 | 0.3 |
| 1.00 | 0.4 | 0.2 | 0.5 | 0.9 | 0.0 |
| 3.00 | 0.1 | 0.1 | 0.0 | 0.8 | 0.8 |
| 6.00 | 0.1 | 0.5 | 0.1 | 0.8 | 0.8 |
| 9.00 | 0.0 | 0.5 | 0.2 | 1.0 | 0.7 |

(b) SRA results for QUAIL (MWEM)

| epsilons | bank | car | shopping | mushroom | adult |
|---|---|---|---|---|---|
| 0.01 | 0.6 | 0.7 | 1.0 | 1.0 | 0.6 |
| 0.10 | 0.8 | 0.5 | 1.0 | 0.4 | 0.4 |
| 0.50 | 0.2 | 0.6 | 0.9 | 0.4 | 0.2 |
| 1.00 | 0.5 | 0.5 | 0.6 | 0.5 | 0.6 |
| 3.00 | 0.5 | 0.5 | 0.3 | 0.4 | 0.6 |
| 6.00 | 0.7 | 0.5 | 0.8 | 0.3 | 0.7 |
| 9.00 | 0.7 | 0.4 | 0.2 | 0.4 | 0.4 |

(a) SRA results for QUAIL (PATEGAN)

| epsilons | bank | car | shopping | mushroom | adult |
|---|---|---|---|---|---|
| 0.01 | 0.6 | 0.3 | 0.7 | 0.3 | 0.8 |
| 0.10 | 0.4 | 0.5 | 0.7 | 0.4 | 0.6 |
| 0.50 | 0.2 | 0.5 | 0.7 | 0.3 | 0.6 |
| 1.00 | 0.7 | 0.5 | 0.8 | 0.3 | 0.2 |
| 3.00 | 0.4 | 0.4 | 0.2 | 0.3 | 0.5 |
| 6.00 | 0.2 | 0.5 | 0.3 | 0.4 | 0.0 |
| 9.00 | 0.6 | 0.5 | 0.6 | 0.4 | 0.4 |

(b) SRA results for QUAIL (DPGAN)

| epsilons | bank | car | shopping | mushroom | adult |
|---|---|---|---|---|---|
| 0.01 | 0.5 | 0.7 | 0.9 | 1.0 | 0.2 |
| 0.10 | 0.6 | 0.6 | 0.5 | 0.9 | 0.6 |
| 0.50 | 0.2 | 0.5 | 0.5 | 0.5 | 0.3 |
| 1.00 | 0.5 | 0.6 | 0.9 | 0.3 | 0.6 |
| 3.00 | 0.6 | 0.5 | 0.2 | 0.5 | 0.5 |
| 6.00 | 0.6 | 0.5 | 0.3 | 0.5 | 0.8 |
| 9.00 | 0.6 | 0.5 | 0.0 | 0.5 | 0.2 |

```
log_frequency=True,
disabled_dp=False,
target_delta=None,
sigma = 5,
max_per_sample_grad_norm=1.0,
verbose=True,
loss = 'cross_entropy',
binary=False,
batch_size = 500,
teacher_iters = 5,
student_iters = 5,
delta = 1e-5
```

DPGAN
```
binary=False,
latent_dim=64,
batch_size=64,
epochs=1000,
delta=1e-5
```

(a) SRA results for QUAIL (PATECTGAN)

| epsilons | bank | car | shopping | mushroom | adult |
|---|---|---|---|---|---|
| 0.01 | 0.9 | 0.3 | 0.6 | 0.3 | 0.5 |
| 0.10 | 0.4 | 0.6 | 0.9 | 0.3 | 0.9 |
| 0.50 | 0.7 | 0.5 | 0.5 | 0.3 | 0.6 |
| 1.00 | 0.0 | 0.5 | 0.9 | 0.3 | 0.9 |
| 3.00 | 0.6 | 0.4 | 0.1 | 0.3 | 0.7 |
| 6.00 | 0.7 | 0.4 | 0.8 | 0.4 | 0.9 |
| 9.00 | 0.7 | 0.4 | 0.9 | 0.3 | 0.8 |

(b) SRA results for QUAIL (DPCTGAN)

| epsilons | bank | car | shopping | mushroom | adult |
|---|---|---|---|---|---|
| 0.01 | 0.8 | 0.2 | 0.9 | 0.5 | 0.2 |
| 0.10 | 0.6 | 0.5 | 0.8 | 0.3 | 0.3 |
| 0.50 | 0.1 | 0.4 | 0.7 | 0.3 | 0.4 |
| 1.00 | 0.7 | 0.5 | 0.5 | 0.3 | 0.4 |
| 3.00 | 0.1 | 0.4 | 0.8 | 0.4 | 0.5 |
| 6.00 | 0.0 | 0.4 | 0.7 | 0.3 | 0.1 |
| 9.00 | 0.3 | 0.4 | 0.1 | 0.5 | 0.1 |

PATEGAN

```
binary=False ,
latent_dim =64 ,
batch_size =64 ,
teacher_iters =5 ,
student_iters =5 ,
delta =1e −5
```

