# OpenReview forum: "Differentially Private Synthetic Data: Applied Evaluations and Enhancements"
_ICLR.cc/2021/Conference — Reject_

### Official Review · AnonReviewer1 · 2020-10-27

**Rating:** 4
**Confidence:** 4

**Review:**

The paper proposes QUAIL, an ensemble of a generative model and a classifier, where both are trained with differential privacy, in order to generate a differentially private dataset (from the generative model) and a label vector (from the classifier). The paper further compared QUAIL with "other" differentially private generative models based on conditional GAN.

My main concern with the paper is that there is no clear contribution. It is hard to judge whether the main claim is that the paper presents a survey of the current differentially private generative models, in which case, the survey part is very short and not in-depth. Or is that the the paper proposes QUAIL, which in many cases performs worse than other generative models. My some other concerns are detailed below:

- There is no mention of \delta used for (\epsilon,\delta) - differential privacy
- Please use something else than \delta for difference between performance measures as it can get confusing
- As the version of GAN used in the paper is conditional, shouldn't the generator be differentially private as well? i.e. how are we protecting the privacy of the labels?
- There is a mention of PATECTGAN performing better even compared to the non-noisy model trained on real data, how is this possible, please add some explanation.

---

> ### Author Response · Authors · 2020-11-18
> **Response to Reviewer 1**
>
> Thank you for the constructive feedback! We address your questions and concerns in the following comments.
>
> Overall Author Comments:
> Thank you for reviewing our paper and providing feedback.
>
> We believe that we provide two clear contributions: the QUAIL hybrid method and a survey of existing state of the art DP techniques (PATE and DPSGD) when applied to an extensive array of scenarios (5 publicly available datasets, 2 large internal datasets, on wide range of \epsilon budget).
>
> We note that QUAIL should not be directly compared to existing methods in that it is an enhancement, not a standalone method, and so can be applied to any existing DP synthetic data method in conjunction with any DP supervised learning method.
>
> Specific Responses:
>
> Reviewer comment: “There is no mention of \delta used for (\epsilon,\delta) - differential privacy.”
>
> Authors: It is true that we failed to highlight the \delta used in the main body of the paper. We do, however, list our model parameters in the appendix,  including our \delta (“target_delta”) parameters, which were all set at 1e-05. We will highlight this accordingly in our final revision.
>
>
> Reviewer comment: “Please use something else than \delta for difference between performance measures as it can get confusing”
>
> Authors: This is a good point, and we will update our figures to use something else to express the difference in our final revision.
>
>
> Reviewer comment: “As the version of GAN used in the paper is conditional, shouldn't the generator be differentially private as well? i.e. how are we protecting the privacy of the labels”
>
> Authors: Based on the post-processing property that any randomized mapping of a differentially private output, is also differentially private, the generator is guaranteed to be differentially private when the generator is trained to maximize the probability of D(G(z)) .  In CTGAN, the authors add the cross-entropy loss between conditional vector and produced set of one-hot discrete vectors into the generator loss. To guarantee differential privacy with the generator, we removed the cross-entropy loss when calculating generator loss.  Thus, the generator is differentially private as well.
>
> Reviewer comment: “There is a mention of PATECTGAN performing better even compared to the non-noisy model trained on real data, how is this possible, please add some explanation.”
>
> Authors: Training supervised learning models on fully synthesized datasets has been observed to at times outperform the same supervised learning model on the real data. Dwork et al. suggested that it can reduce overfitting. They said: “The intuition is that if we can learn about the data set in aggregate while provably learning very little about any individual data element, then we can control the information leaked and thus prevent overfitting.“ Furthermore, in line with other recent literature, we concluded that, with fine tuning, high-quality synthetic data (like that generated by PATECTGAN at higher epsilons) can be more useful for pre-training than the real data that may not contain a more optimal distribution of edge cases. We will include a note on this in our revision – and in practice we often observe this phenomena.
>
> (See Dwork, Cynthia, et al. "The reusable holdout: Preserving validity in adaptive data analysis." Science 349.6248 (2015): 636-638. See An Annotation Saved is an Annotation Earned: Using Fully Synthetic Training for Object Instance Detection, https://arxiv.org/abs/1902.09967. See McCormac et al. SceneNet RGB-D: Can 5M Synthetic Images Beat Generic ImageNet Pre-Training on Indoor)

---

> > ### Comment · AnonReviewer1 · 2020-11-22
> > **response**
> >
> > Thank you for the detailed response. My first comment still stands as I do find that the paper does not provide clear enough background for the survey, I would have preferred if the paper focused on QUAIL solely, as this would have provided more space for exploring QUAIL in depth, which the paper lacks currently.
> >
> > That space could have been further used to explore the regression results where RMSE of DP models is lower than non-private model. I do agree that DP methods can SOMETIME outperform non-DP methods, but that is rare and happens usually when the non-DP model has room for improvement (such as better control for overfitting to improve generalization), and is often seen in "generic" DP models, I find it peculiar seeing it for DP synthetic data, which already is a noisy version of real dataset.
> >
> > Another observation: In figures 6 and 7, it seems like the epsilon split doesnt matter that much, as the difference between vanilla and quail is pretty small for many data sizes, any explanation for that?

---

> > > ### Author Response · Authors · 2020-11-24
> > > **Response**
> > >
> > > "My first comment still stands..."
> > >
> > > Point taken, we appreciate the advice about how we could structure the paper or papers for more impact!
> > >
> > >
> > > "I do agree that DP methods can SOMETIME outperform non-DP methods..."
> > >
> > > It is true that there are sometimes cases where naïve use of holdouts, or improper regularization, causes vanilla regression to perform worse than differential privacy.  Another way to look at this is that differential privacy gives some degree of regularization "for free", in settings where privacy is important.
> > >
> > > In practice, we have observed generalization benefits in 3 scenarios: 1) differential privacy applied to aggregate features before training a model, 2) differential privacy applied during the training process (e.g. DP-SGD), and 3) differentially private synthetic data.  Note that #3, differentially private synthetic data, is the setting of the Dwork/Hardt paper on reusable synthetic holdouts (the "Thresholdout" algorithm is interactive in that setting, but the same ideas apply).
> > >
> > > We feel that this is an underexplored area of research.
> > >
> > >
> > > "In figures 6 and 7, it seems like..."
> > >
> > > The simplest explanation is that the epsilon budget (3.0) was more than sufficient in training the supervised learning model for this specific task. As we showed in some of the experimentation added as part of our revision, there are some scenarios where the embedded DP learning model, with a fraction of epsilon used for the vanilla scenario, performs almost as well as the vanilla scenario. In this case, we believe QUAIL can help allocate excess epsilon.

---

### Official Review · AnonReviewer2 · 2020-10-30
**Need some improvement**

**Rating:** 4
**Confidence:** 3

**Review:**

The authors proposed QUAIL, an algorithm that uses a supervised model and a synthetic data model to generate synthetic data that is good for downstream tasks. It also shows some empirical evaluations of the algorithm.

The technical part, especially the experiments, might need some improvement. The main issue is that we cannot draw a clear conclusion from the experimental results. I would suggest taking one or two epsilon values, and look at the results in more detail to see if we can find any trend. Also, generating synthetic data under differential privacy is not an easy task, so I think it's ok to skip the regime where epsilon < 1, or try even larger epsilon to get a reasonable utility first.
However, I would like to say that I very much appreciate the authors' effort in conducting experiments on quite a few datasets, and their integrity in presenting all the results no matter positive or negative.

The paper can be better organized, for example,
- The notations in the algorithm should be clearly explained/defined before the algorithm (for example, N and X), and some intuition can be added after the algorithm description. In the algorithm description itself, I think maybe it's clearer to defined the target dimension and the rest of the dimension separately, i.e. defining one sample as (feature, target) and later on we will have (synthetic feature, synthetic target). In the "split" part, I guess eps_C should be (1-p)*eps.
- The part below Theorem 3.1 might better be put into the experiment section than the algorithm section.
- In quite a few figures, we see interesting trends like some algorithm can have worse utility as epsilon grows. So it might be important to report the standard deviation of the algorithm for readers to better understand what was going on. Also, the texts in the figures can be made larger.
- The paper called the algorithm "ensemble method". I feel like ensemble means something specific in ML, and simply using two different models together doesn't quite seem like ensemble. Maybe I'm understanding something here but it should be better explained.

---

> ### Author Response · Authors · 2020-11-18
> **Response to Reviewer 2**
>
> Thank you for the constructive feedback! We address your questions and concerns in the following comments.
>
> Overall Author Comments:
> Thank you for expressing your appreciation for our “integrity in presenting all the results no matter positive or negative” and for our “effort in conducting experiments on quite a few datasets.” We were motivated to conduct our extensive experiments and represent them faithfully after we observed a lack of similarly broad, candid experimentation on synthetic data in current literature.
>
> Specific Responses:
>
> Reviewer comment: “The main issue is that we cannot draw a clear conclusion from the experimental results. I would suggest taking one or two epsilon values, and look at the results in more detail to see if we can find any trend.”
>
> Authors: This is a useful suggestion. We have further investigated the epsilon value of 3.0 in our revision, especially in trying to peel back the QUAIL method to produce some justification for why it works. We have included our justification here, and new figures in the paper body:
> “By first assessing the TSNE clustering in Figures 8 and 9, we see that not only is the synthetic data produced by QUAIL very similar to the real data, but the accuracy of the labeling for the embedded model (in this case, DPLR) is also very similar. Further investigation into data scale revealed that the QUAIL method takes advantage of allocating excess epsilon when datasets are large. As datascale increases, the sensitivity of the differentially private model decreases and so less epsilon can be used more efficiently. Thus, we see that an exaggerated difference between DPLR embedded in QUAIL (with an epsilon of 2.4) and DPLR with an epsilon of 3.0 for a dataset of 20,000 samples. In this case, the embedded DPLR model accuracy suffers, and so does the learning utility of the produced synthetic data. Conversely, as we increase the data size to 50,000 and 100,000 samples, we see that the internal model (with epsilon 2.4) can match the performance of the vanilla model (with epsilon 3.0). Then, the synthetic dataset serves only to augment the performance by small but significant margin (in Figure 10, we see a bump of three percent to f1 score.”
>
>
> Reviewer comment: “The notations in the algorithm should be clearly explained/defined before the algorithm (for example, N and X), and some intuition can be added after the algorithm description. In the algorithm description itself, I think maybe it's clearer to defined the target dimension and the rest of the dimension separately, i.e. defining one sample as (feature, target) and later on we will have (synthetic feature, synthetic target). In the "split" part, I guess eps_C should be (1-p)*eps.”
>
> Authors: Thank you for the notes on the notation/format of the QUAIL algorithm. We have made them in our revision. If the review could clarify what they meant by “, i.e. defining one sample as (feature, target) and later on we will have (synthetic feature, synthetic target),” we can complete the changes for our final revision.
>
>
> Reviewer comment: “The part below Theorem 3.1 might better be put into the experiment section than the algorithm section”
>
> Authors: Thank you, we have moved this in our revision.
>
>
> Reviewer comment: “In quite a few figures, we see interesting trends like some algorithm can have worse utility as epsilon grows. So it might be important to report the standard deviation of the algorithm for readers to better understand what was going on. Also, the texts in the figures can be made larger”
>
> Authors: Thank you! In our final revision, we will add standard deviations in our figures, and increase the text size.
>
>
> Reviewer comment: “The paper called the algorithm "ensemble method". I feel like ensemble means something specific in ML, and simply using two different models together doesn't quite seem like ensemble. Maybe I'm understanding something here but it should be better explained”
>
> Authors: This is a good point – your understanding is correct. It is slightly misleading to describe QUAIL as an “ensemble” method between two models as it doesn’t do any traditional boosting, bagging, etc. In our revision, we have reworded and removed references to QUAIL as an ensemble, replacing it with “hybrid approach.”

---

### Official Review · AnonReviewer3 · 2020-11-02
**The idea is interesting and simple whereas no clear reason that the proposed method works well is not explained.**

**Rating:** 4
**Confidence:** 4

**Review:**

The proposed method works as follows. Given samples are partitioned into two parts; one is for classifier training and the other is for data synthesizer training. Both are trained in a differentially private manner. After training, the DP synthesizer generates samples and the DP classifier labels them so that the resulting samples can be used as training samples. By the post-processing theorems, the resulting are differentially private, which are published as synthesized samples.

The idea is interesting, simple, and unique. Also, experimental results demonstrate that the  models trained with the proposed method give s better F1 score compared to the existing methods. One limitation of this manuscript is that the reason why the proposed scheme can give better classification accuracy is not discussed. Also, the reason why the RMSE of the regression model trained with this scheme is worse than other methods is not examined, either. One quick thought is that the proposed scheme preserves the cluster structure of the samples well and therefore the classification model trained with the resulting sample has good accuracy. In contrast, the metric structure behind the samples is not preserved well and therefore the regression model does not have good RMSE. I am not sure this is correct or not, but anyway, I think further consideration on these issues will be interesting and needed for this type of experimental study to find a clue to improve data synthesization with DP guarantee.

Minor:
FIg 6 is in the Appendix, not in the main body. Also, many important claims (mainly in experimental results) are given with results in the Appendix. The main claim should be constructed with the contents in the main body.

---

> ### Author Response · Authors · 2020-11-18
> **Response to Reviewer 3**
>
> Thank you for the constructive feedback! We address your questions and concerns in the following comments.
>
> Overall Author Comments:
> We appreciate your interest in our proposed method. We encourage you to consider that our extensive experiments, especially in applied scenarios, with existing DP synthetic data (and modifications to existing methods) represent a valuable secondary contribution.
>
> Specific Responses:
>
> Reviewer comment: “One limitation of this manuscript is that the reason why the proposed scheme can give better classification accuracy is not discussed… One quick thought is that the proposed scheme preserves the cluster structure of the samples well and therefore the classification model trained with the resulting sample has good accuracy.”
>
> Authors: This is a useful suggestion, and something had explored in our experimentation. We have included an analysis of the “cluster structure” as part of our revision.
>
>
> Reviewer comment: “Also, the reason why the RMSE of the regression model trained with this scheme is worse than other methods is not examined, either… In contrast, the metric structure behind the samples is not preserved well and therefore the regression model does not have good RMSE.”
>
> Authors: We believe we addressed this concern on page 7, in section 5. We wrote “For QUAIL-enhanced models, the RMSE is considerably larger than the real and other DP synthetic data. We attribute this to a weakness of the embedded regression model (DP Linear Regression) in QUAIL for this data scenario.” We appreciate the reviewer’s thoughts, and believe there could be some validity to their statement. However, we are confident in our assertion that the weakness of the DP Linear Regression model embedded in QUAIL is the root of the overall poor performance. We present results that suggest a more robust DP Linear Regression model embedded in QUAIL would lead to improved performance in our revision.
>
>
> Reviewer comment: “Minor: FIg 6 is in the Appendix, not in the main body. Also, many important claims (mainly in experimental results) are given with results in the Appendix. The main claim should be constructed with the contents in the main body. “
>
> Authors: This is a good point. We have move Figures 6 and 7 into the main body, as well as some other pertinent figures in our revision.

---

### Decision · Program_Chairs · 2021-01-07
**Final Decision**

**Decision:**

Reject

**Comment:**

The paper surveys existing differentially private data synthesis
methods, and introduces an algorithm that learns both a generator and
a classifier in a differentially private mode.

The problem is highly timely and important. Results are promising.

Main remaining concerns after discussion between the reviewers and the
authors are:

- reason why the proposed scheme can give better classification
accuracy, should be clarified more

- unclarity on conclusions that can be drawn from the experiments. The
revised version has improved on this somewhat.

One explanation for the problems was suggested to be that the paper
tries, at the same time, to both present a new method and be a
survey. Is hard to do in a short paper, and as a result, the paper
lacks focus. At the very least, more work is needed.

The authors are encouraged to continue their work on this
important problem, and the review comments hopefully help in that.